# Exploring and Addressing Reward Confusion in Offline Preference Learning

## Abstract

Spurious correlations in a reward model's training data can prevent Reinforcement Learning from Human Feedback (RLHF) from identifying the desired goal and induce unwanted behaviors. In this work, we study the reward confusion problem in offline RLHF where spurious correlations exist in data. We create a lightweight benchmark to study this problem and propose a method that can reduce reward confusion by leveraging model uncertainty and the transitivity of preferences with active learning.

## 1  Introduction

For many real-world tasks, designing adequate reward functions is challenging, which has led to the rise of Reinforcement Learning from Human Feedback (RLHF) [6]. In this work, we study a failure mode of offline RLHF that we refer to as *reward confusion*. This occurs when the reward $R$ in a Markov Decision Process (MDP) is a function of features $z_1, \ldots, z_n$ inferred from the observation-action pair $(o, a)$. In a simplified scenario, $R$ relies on $z_1$ but not $z_2$, yet $z_1$ and $z_2$ are highly correlated in the training data. An empirical risk minimizer might mistakenly conclude that $z_2$ affects $R$. As we'll see, this incorrect dependence can lead to failures when training a policy against the learned reward function. We graphically illustrate this problem in Figure 1.

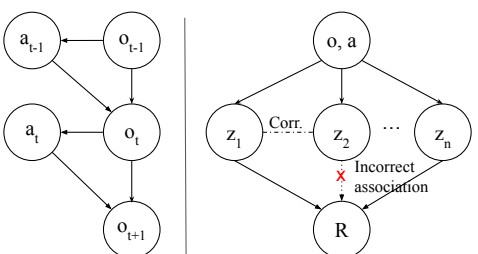

Figure 1: Illustration of a simplified MDP (left) and reward confusion (right). The reward $R$ is a function of the feature $z_1$, but not $z_2$. Spurious correlation between $z_1$ and $z_2$ can cause a network to wrongly model $R$ as a function of $z_2$.

To better understand this phenomenon, we created a benchmark environment called *Confusing Minigrid (CMG)* that tests reward confusion in models. We carefully designed six tasks with three types of spurious information for the minigrid environment, which we introduce in detail in Appendix A. We will open source the benchmark's code soon.

Besides the CMG benchmark, one other our major contributions is an algorithm named Information-Guided Preference Chain (IMPEC) designed to address the reward confusion problem. It involves two stages of training: First, we use information gain as the acquisition function to select comparison rollouts that reduce uncertainty about the reward function. Second, we form a complete preference ordering over the set of selected rollouts, rather than just a partial ordering as in traditional RLHF.

Our experiments show that these techniques together improve sample efficiency while reducing reward confusion. We show in Section 4 that using the same comparison budget, IMPEC can

Submitted to Workshop on Bayesian Decision-making and Uncertainty, 38th Conference on Neural Information Processing Systems (BDU at NeurIPS 2024). Do not distribute.

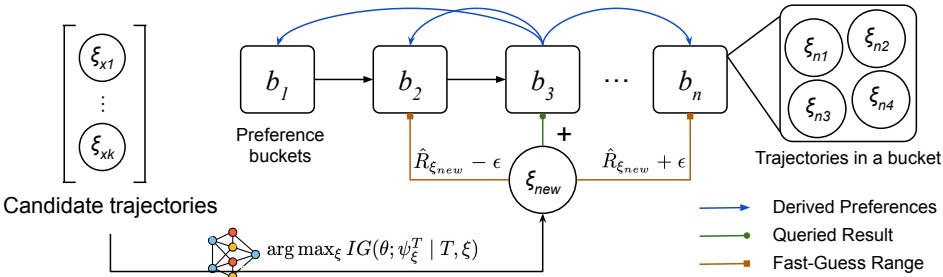

Figure 2: The IMPEC algorithm creates a sorted preference chain of $n$ buckets, each containing one or more rollouts with equal returns.

outperform many other active preference learning baselines. To the best of our knowledge, it is the first algorithm that attempts to solve the reward confusion problem in preference learning.

## 2   Related Work

**Causal Confusion**   The problem of *causal confusion*, which refers to models learning to depend on spurious correlations in the training data, has been studied in behavioral cloning [7], reinforcement learning [12], and reward learning [20]. Past work shows empirically and theoretically that spurious correlations and confounders in the training set can worsen an agent's deployment performance [22, 9]. Reward confusion is essentially causal confusion that occurs during reward learning.

**Goal Misgeneralization**   While past work on causal confusion studies it as a cause of complete failure to learn goal-directed behavior, it can also make agents optimize for incorrect goals, i.e. goal misgeneralization. For example, in Procgen's CoinRun, the coin to be picked up is always on the right. RL agents can confuse "running to the right" with the real goal of "getting the coin" [10]. Generally, a model's behavior can be consistent with a goal, but it may not be the test-time goal [16].

**Preference Learning**   Learning reward models from preference labels [6] have gained traction due to their low cost compared to expert demonstrations  [23] or language inputs [21]. We have also seen progress in other tasks of "reward engineering", e.g. reward hacking [18].

## 3   Method

**Models**   We consider an agent in an environment following the Markov Decision Process (MDP) defined by $(\mathcal{S}, \mathcal{A}, \mathcal{P}, \mathcal{R})$. $\mathcal{S}$ is the state space, $\mathcal{A}$ is the action space, $p : \mathcal{S} \times \mathcal{S} \times \mathcal{A} \to [0, \infty)$ is the transition probability density. A rollout $\xi = (s_t, a_t)$ is a sequence of states and actions. Given unranked rollouts $\Xi$, our algorithm actively collects ranking information to sort them into an ordered list $T = \langle \xi_1, \xi_2, ..., \xi_n \rangle$. The rank of rollout $\xi \in T$ is denoted by $\psi_\xi^T$. On each transition, the environment emits a reward $\mathcal{R} : \mathcal{S} \times \mathcal{A} \to \mathcal{R}$. Our goal is to obtain $\mathcal{R}^*$ that induces correct policies.

**Preferences**   We model the human's probability of preferring $\xi_1$ in a pair $(\xi_1, \xi_2)$ through the Shepard-Luce choice rule [17, 13]: $P[\xi_1 \succ \xi_2] = \frac{\exp \sum_t r(o_t^1, a_t^1)}{\exp \sum_t r(o_t^1, a_t^1) + \exp \sum_t r(o_t^2, a_t^2)}$. We extend this model to a ternary one by allowing the human to flag when two rollouts are equally good, $\xi_1 \equiv \xi_2$. We use cross-entropy loss to improve reward model's predictions for human's true preference.

### 3.1   Information-Guided Preference Chain (IMPEC)

**Key intuition: Increase Contrast Among Valuable Rollouts.**   In most preference comparison algorithms, a rollout $\xi_1$'s relation is considered explicitly only with another one $\xi_2$. Suppose that in the ground truth, $\xi_1 \succ \xi_2 \succ \xi_3 \succ \xi_4$, and we already know $\xi_1 \succ \xi_2, \xi_3 \succ \xi_4$. To figure out $\xi_1$ and $\xi_2$'s relationship with $\xi_3$ and $\xi_4$, the most efficient query is whether $\xi_2 \succ \xi_3$. Once we establish that, we can immediately obtain the preference relations on all four rollouts.

**Creating and Maintaining a Preference Chain**   We maintain an ordered chain for rollouts. Starting from an empty chain, for each new rollout we queried from the dataset, we imitate insertion sort by recursively finding the ranking of it using human's preference labels. Hence, by the time we observe

all the rollouts, we have a sorted list of rollouts, ordered according to human preferences. Rollouts can have identical returns, so we treat each element of the chain as a *bucket* $b \in \mathcal{B}$ of rollouts with the same return. If the human decides that a new rollout $\xi_{\text{new}}$ is equally preferred to $\xi_m$ in bucket $b_m$, then $\xi_{\text{new}}$ will be added to $b_m$. On the other hand, if $\xi_{\text{new}} \succ \xi_m$ and $\xi_{\text{new}} \prec \xi_{m-1}$ ($\xi_{m-1}$ resides in a previous bucket $b_{m-1}$), then the algorithm will insert a new bucket containing only $\xi_{\text{new}}$ in between $b_m$ and $b_{m-1}$. This ensures that $b_0$ contains the best rollouts seen so far and $b_n$ contains the least preferred rollouts (where $n$ is the chain length). We illustrate this process in Figure 2.

Our reward model is a Bayesian neural network (BNN) [3] which maintains a Gaussian distribution over a network's weights and biases. As we will see, this allows us to incorporate epistemic uncertainty over reward functions into the active selection procedure. In the noiseless case, insertion sort needs $O(\log n)$ queries to find the position for $\xi_{\text{new}}$. However, we have access to a partially trained reward network, which we use to guess the rank for $\xi_{\text{new}}$, reducing the number of buckets we must search over. We include more design details in Appendix B for the design of the fast query.

**Information Gain** Given an existing chain of rollouts, we use information gain as the acquisition function to decide which rollouts to compare next, so we reduce the most uncertainty over network weights. The information gain over network weights $\theta$ by selecting a rollout $\xi \in \Xi$ for ranking is

---

**Algorithm 1** The IMPEC Algorithm

**Require:** Preference dataset D, network $\theta$, query budget Q
  $T \leftarrow []$
  **while** not converged **do**
    $\theta \leftarrow \text{SupervisedTrain}(\theta, D)$
    **if** budget not reached **then**
      $\xi \leftarrow \arg\max_\xi I(\theta; \psi_\xi^T \mid T, \xi)$
      $\psi_\xi^{T*} \leftarrow \text{InsertionSort}(\xi, T, \theta)$
      $T \leftarrow T \cup \xi$
      $D \leftarrow D \cup \text{DerivePreferences}(\xi, T, \psi_\xi^T)$
    **end if**
    $i \leftarrow i + 1$
  **end while**

---

$$I\left(\theta; \psi_\xi^T \mid T, \xi\right) = H(\theta \mid T, \xi) - H\left(\theta \mid \psi_\xi^T, T, \xi\right) \tag{1}$$

where $\psi_\xi^T$ is the rollout's ranking on chain $T$. Intuitively, it measures how much we expect to reduce uncertainty about the weights after observing the ranking $\psi_\xi^T$ of rollout $\xi$. As shown in Appendix C, Equation 1 (information gain) can be approximated by drawing $M$ weight samples, $\theta_1, \theta_2, \ldots, \theta_M \sim \theta$, from the posterior through

$$\frac{1}{M} \sum_{i=1}^{M} \sum_\psi P\left(\psi_\xi^T \mid T, \theta_i, \xi\right) \cdot \log\left(\frac{M \cdot P\left(\psi_\xi^T \mid T, \theta_i, \xi\right)}{\sum_{\theta_j} P\left(\psi_\xi^T \mid T, \theta_j, \xi\right)}\right) \tag{2}$$

$P\left(\psi_\xi^T \mid T, \theta, \xi\right)$ is a complicated distribution, and so we (loosely) approximate it with Equation 3. Intuitively, it is proportional to the probability that $\xi_i \succ \xi \succ \xi_{i+1}$.

$$P\left(\psi_\xi^T = i \mid T, \theta, \xi\right) \propto P\left(\xi_i \succ \xi \mid \theta\right) \cdot P\left(\xi \succ \xi_{i+1} \mid \theta\right) \tag{3}$$

We summarize the complete process in Algorithm 1. The network is first supervised trained on the preference dataset $D$ using cross entropy loss with $P[\xi_1 \succ \xi_2]$ modeled through the Shepard-Luce choice rule. With the limited query budget for human preferences, we first find out the rollout $\xi$ whose ranking $\psi_\xi^T$ on chain $T$ will provide the most information gain over the model weights $\theta$. Then we use insertion sort to find out $\xi$'s real ranking $\psi_\xi^{T*}$ in the chain. We add the rollout $\xi$ onto the appropriate position of the chain $T$, then based on its position, derive preference labels with all other rollouts on the chain. We repeat this process until the network weight has converged.

## 4 Experiments

**Experiment Settings** We compare our method with the standard RLHF algorithm, and two other RLHF with active learning methods: pairwise information gain [2] and pairwise volume removal [14]. The information gain (IG) method is similar to ours but reasons only about individual preference pairs and not about the result of the ranking process. The volume removal method was designed in

|  | **Baseline** | **IMPEC** | **Infogain** | **Vol Removal** |
|---|---|---|---|---|
| Empty | 12.90±8.20 | 18.13±2.15[†] | 7.33±9.98 | 14.66±8.41 |
| Dyn Obs | 7.17±6.56 | 11.78±2.82[†] | 5.34±6.75 | 4.06±6.03 |
| Lava | 8.70±12.83 | 17.65±1.97[†] | 13.33±8.35 | 9.75±9.39 |
| Lava Pos. | 3.51±1.26 | 7.21±2.20[†] | 3.60±1.52 | 4.66±2.53 |
| Fetch | 10.60±2.85 | 11.52±1.17 | 9.93±2.47 | 10.80±1.51 |
| Door | 1.58±0.31 | 1.67±0.53 | 1.55±0.48 | 1.68±0.71 |

Table 1: Ground truth returns (mean ± standard deviation) for different methods on Confusing Minigrid. † indicates a p-value of ≤ 0.1 (vs. baseline).

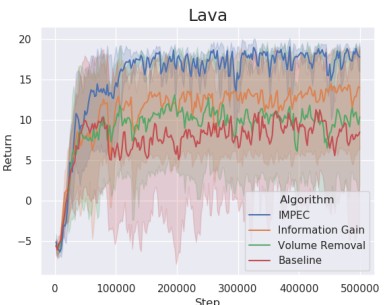

Figure 3: The learning curves for 4 algorithms trained on the Lava task.

the linear reward setting to reduce the volume of weight vectors supported under the posterior after each preference update. We conduct experiments on 6 CMG tasks. Detailed information on each task and their added spurious correlations can be found in Appendix A.

We first perform offline reward learning, and then apply online reinforcement learning using the learned reward function to obtain a policy. The RL agent receives rewards from the learned function instead of the environment, and is trained with Proximal Policy Optimization [15]. More detailed experiment and hyperparameter settings can be found in Appendices D and E.

**Main Results** Performance on all six tasks can be found in Table 1, with each run repeated over five seeds. We further compute the p-values of the results being better than baseline performance, with complete results in Appendix H. Except for the task Go To Door where all algorithms perform poorly, IMPEC has a higher mean return than other algorithms, and often a lower standard deviation. Although IMPEC achieves a higher mean than the other methods on most tasks, its p-values are ≤ 0.1 (per appendix H). This provides some (albeit not particularly strong) statistical evidence that IMPEC has a better mean performance distribution from the baseline. In contrast, the volume removal and information gain methods are often statistically indistinguishable from the baseline.

The decisive factor for each algorithm's performance is how often they fail: Out of the 5 seeds, how often does the reward function learn to optimize for the spurious goal? All methods but IMPEC have fairly high probability of taking the spurious feature as the "correct" feature, and hence rewarding incorrect behaviors and obtaining low ground truth returns. We plot the learning curves for each algorithm in the Lava task, where the baseline curve has the largest standard deviations. We observe similar phenomena in other tasks. We include an ablation study in Appendix F. The complete learning curves are included in Appendix J.

**Limitations and Future Work** A limitation of IMPEC is its potential sensitivity to noise in preferences. In our experiments, we keep a relatively low noise level, and we believe more sophisticated algorithms could improve robustness to noise, perhaps inspired by past work on noisy binary search [8, 5]. Our results suggest a deeper connection between the quality of preference datasets and the efficiency of preference learning algorithms. In appendix G, we show some first steps of a graph theoretic analysis for reasoning about preference dataset quality. We are interested in further exploring the influence of graph-theoretic qualities and their effects on preference learning, and using the insights in future algorithm design.

# 5 Conclusion

This work studies the reward confusion problem. Our experiments on the Confusing Minigrid benchmark show that reward confusion in offline preference learning can lead to undesired policy behaviors. The benchmark is easy to configure, and we expect it to be particularly useful for iterative research. In addition, we proposed IMPEC to reduce the impact of reward confusion. It exploits preference transitivity and obtains decent empirical performance on tasks with different sources of reward confusion. We believe that the findings of our work will be helpful for making AI more aligned with human values.

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

# A Confusing Minigrid Definition

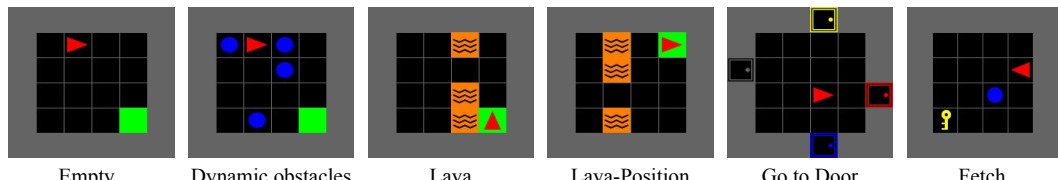

Empty    Dynamic obstacles    Lava    Lava-Position    Go to Door    Fetch

Figure 4: The six Confusing Minigrid tasks. The tasks require agents to move to goal positions, go to doors, or fetch objects.

Reward learning methods are often evaluated on benchmarks originally developed for reinforcement learning algorithms, like Atari [1] or MuJoCo continuous control [19]. These reinforcement learning benchmarks cannot easily be used to test reward confusion failures. Thus we created six new tasks based on Minigrid [4], which together form the Confusing Minigrid benchmark.

| Task name | Extra | Position | Color | DistShift |
|---|---|---|---|---|
| Empty | Y | N* | N* | N |
| Dynamic Obs. | Y | N* | N* | N |
| Lava | Y | N* | N* | N |
| Lava-Position | N* | Y | N* | Y |
| Go to door | N* | Y | N* | Y |
| Fetch | N* | N* | Y | Y |

Table 2: A summary of Confusing Minigrid tasks. Y/N is short for yes/no. An asterisk means that a feature is off by default but can be optionally turned on. The "extra" column refers for having extra observation dimensions for the task. The position and color columns refer to spurious correlations that come from position and color information, respectively. DistShift means there is a difference between the training and testing environments.

## A.1 Spurious Correlations from Extra Observations

This set of tasks tests whether spurious correlations with redundant observation dimensions can interfere with learning. In these tasks, an agent needs to navigate to the goal cell. It can observe the state of a glass of water it is holding, which exhibits "ripples" as it moves, and calms down if the agent's position remains unchanged. On rollouts where the agent moves straight to the goal and stops, the level of ripples will be predictive of whether the agent has reached the goal, even though in general it is possible to cause the ripples to disappear by stopping in any location and not just at the goal. The training and testing variants of these tasks are the same.

**Empty**  This is the simplest task in the benchmark. The agent needs to navigate to the goal cell, with all other cells being empty. The environment gives a positive reward when the agent reaches the goal, and zero otherwise.

**Dynamic Obstacles**  This task augments the Empty task with obstacles that show up randomly in non-goal cells at each time step, which block the agent's path. We include this task because the obstacles may stop the agent at a random grid while it is collecting rollouts, which increases the chance that the reward learning algorithm later realizes that "stabilizing the water" is not the correct goal.

**Lava**  The environment contains a row of lava cells with a gap that allows the agent to cross and reach the goal. Standing on the lava cells gives a reward of -1. We use this task to observe if negative rewards will have any impacts on learning spurious correlations, and if the correlation is exploited, whether the agent will at least avoid the lava (low reward) area.

### A.2  Spurious Correlations from Distributional Shifts

We design these tasks with different training and testing variants. The training environments have a 90% probability where the goal configuration is spurious.

**Lava-Position**  It is a variant of Lava with changing goal positions. In the training variant, the goal cell is usually located at one particular location. In the test variant, the goal grid can appear at other locations too.

**Go to Door**  In this task, an agent is asked to move to a position adjacent to the goal door embedded in one of the four walls surrounding the grid. There are always four doors in the environment, and the goal door is most likely to be placed in the upper wall during training. During testing, the goal door can be placed in any of the four walls.

**Fetch**  The agent's goal in this task is to pick up a key. There is usually a distractor object in the environment that an agent can also pick up. In the training variant of this task, most keys are yellow, and most distractor objects are non-yellow. At the test time, the keys and distractor objects can appear in any color with equal chance.

For all these tasks, the agent receives a reward of $+1$ when the goal condition is satisfied. We summarize the task settings in Table 2. Changing the confounding type in Confusing Minigrid is as easy as modifying a keyword argument when initializing the environment. Training a standard preference learning algorithm on the tasks with the simplest observation type takes around 2.5 hours on a single Nvidia A6000 GPU, which is faster than many other image-based environments or complex control tasks.

## B  Use a Partially Trained Model to Reduce Queries

The network will first update its reward predictions on rollouts in each bucket $\hat{R}_{b_i}$, then predict the incoming rollout's reward $\hat{R}_{\xi_{\text{new}}}$. Instead of using a point estimate $\hat{R}_{\xi_{\text{new}}}$, we use an interval $[\hat{R}_{\xi_{\text{new}}} - \epsilon, \hat{R}_{\xi_{\text{new}}} + \epsilon]$ to fast-guess where $\xi_{\text{new}}$ may belong. IMPEC queries human preferences of two pairs $(\xi_{\text{new}}, \xi_l)$ and $(\xi_{\text{new}}, \xi_u)$. $\xi_l$ and $\xi_u$ are the rollouts that are closest to the prediction lower bound $\hat{R}_{\xi_{\text{new}}} - \epsilon$ and the upper bound $\hat{R}_{\xi_{\text{new}}} + \epsilon$, respectively. The $\epsilon$ we use is the standard deviation of $\hat{R}_{\xi_{\text{new}}}$ by $M$ samples of the BNN. After $\xi_{\text{new}}$ is added into the chain, we can derive preference relations between it and other rollouts by transitivity.

## C The Information Gain Objective Derivation

This derivation is adapted from [2].

$$I(\theta; \psi_\xi^T \mid T, \xi) = H(\theta|T,\xi) - H(\theta|\psi_\xi^T, T, \xi)$$

$$= -\mathbb{E}_{\theta,T,\xi}\left[\log P(\theta|T,\xi)\right] + \mathbb{E}_{\theta,\psi_\xi^T,\xi,T}\left[\log P(\theta|\psi_\xi^T, T, \xi)\right]$$

$$= -\mathbb{E}_{\theta,\psi_\xi^T,\xi,T}\left[\log P(\theta|T,\xi)\right] + \mathbb{E}_{\theta,\psi_\xi^T,\xi,T}\left[\log P(\theta|\psi_\xi^T, T, \xi)\right]$$

$$= \mathbb{E}_{\theta,\psi_\xi^T,\xi,T}\left[\log P(\theta|\psi_\xi^T, T, \xi) - \log P(\theta|T,\xi)\right]$$

$$= \mathbb{E}_{\theta,\psi_\xi^T,\xi,T}\left[\log \frac{P(\psi_\tau^{T_n}|T,\theta,\xi)P(T,\theta,\xi)}{P(\psi_\xi^T, T, \xi)} - \log\frac{P(\theta,T,\xi)}{P(T,\xi)}\right]$$

$$= \mathbb{E}_{\theta,\psi_\xi^T,\xi,T}\left[\log \frac{P(\psi_\xi^T|T,\theta,\xi)P(T,\theta,\xi)}{P(\psi_\xi^T|T,\xi)P(T,\xi)} - \log\frac{P(\theta,T,\xi)}{P(T,\xi)}\right]$$

$$= \mathbb{E}_{\theta,\psi_\xi^T,\xi,T}\left[\log \frac{P(\psi_\xi^T|T,\theta,\xi)}{P(\psi_\xi^T|T,\xi)}\right]$$

Evaluating this expression requires us to compute a conditional probability with respect to $\theta$, which is a random variable capturing our current uncertainty over the reward network weights. We can approximate the distribution $p(\theta)$ by sampling $M$ weights $\theta_1, \theta_2, \ldots, \theta_M \sim p(\theta)$ and then treating $\theta$ as if it were a uniform mixture over the samples; i.e.

$$p(\theta) \approx \frac{1}{M}\sum_{i=1}^M \delta_{\theta=\theta_i},$$

where $\delta_{\theta=\theta_i}$ denotes a Dirac distribution at $\theta_i$. Using this approximation, our mutual information becomes:

$$I(\theta; \psi_\xi^T \mid T, \xi) \approx \mathbb{E}_{\theta,\psi_\xi^T,\xi,T}\left[\log \frac{P(\psi_\xi^T|T,\theta,\xi)}{\frac{1}{M}\sum_{j=1}^M P(\psi_\xi^T|T,\theta_j,\xi)}\right]$$

$$= \mathbb{E}_{\theta,\psi_\xi^T,\xi,T}\left[\log \frac{M \cdot P(\psi_\xi^T|T,\theta,\xi)}{\sum_{j=1}^M P(\psi_\xi^T|T,\theta_j,\xi)}\right]$$

$$\approx \frac{1}{M}\sum_{i=1}^M \sum_{\psi_\xi^T} P(\psi_\xi^T|T,\theta_i,\xi) \cdot \log \frac{M \cdot P(\psi_\xi^T|T,\theta_i,\xi)}{\sum_{j=1}^M P(\psi_\xi^T|T,\theta_j,\xi)}$$

## D Detailed Experiment Settings

**Offline Dataset and Tasks**  In real-world settings where failures are much more costly than minor failures, rollout datasets will be skewed towards higher-return rollouts. We emulate this by constraining the number of rollouts in the low reward region (return $\leq 5$) to be at most 10% of the dataset.

**Query and Data Budgets**  Preference comparison algorithms typically obtain $n$ pairs of (query, label), $[(\xi_{i1}, \xi_{i2}), \text{label}_i]_{i=1}^n$ for binary classification. For simpler tasks (Empty, DynObs, Lava, and Fetch), we set the query budget to be 300. That is, baselines have access to 300 (query, label) pairs, $[(\xi_{i1}, \xi_{i2}), \text{label}_i]_{i=1}^{300}$. IMPEC requires additional queries to precisely rank each sampled rollout within the candidate list, so we constrain it to use 150 pairs $[(\xi_{i1}, \xi_{i2}), \text{label}_i]_{i=1}^{150}$, and use the remaining 150 query budget to perform insertion sort for a selected subset of rollouts (decided by IG). For the harder tasks (Go to Door and Lava-Position), all algorithms are given a budget of 600 (query, label) pairs. IMPEC can access 400 data pairs, and use the remaining 200 query budget for sorting.

**Dataset Creation** For each task, we first train an RL agent to the expert level, saving its policies at various timesteps. We then take 3 of its policy snapshots - an almost random policy, an expert policy, and one in-between to generate rollouts. The number of rollouts falling within the low-reward ($\leq 5$) region are controlled to take up within 10% of the dataset.

**IMPEC Training** We typically train an algorithm with 20 epochs. For IMPEC, we evenly divide its query budget from epoch 1 to epoch 15, using up all queries in this period and ranking as much uncertain rollouts as possible. We then use the remaining 5 epochs for learning the full dataset. After sampling the initial preference pairs, IMPEC will not obtain new rollouts from the dataset, and perform learning only by querying humans for ranking the given rollouts.

**Environment Observations** The environment observations are in the vector form, which contains [agent position, agent direction, special grid info]. The special grid can be the goal/lava/door, etc., and its information is an encoding of its grid type, current position, and color.

**Ablation Studies** In the "no active learning" experiment, we turn off the active selection function, and randomly pick rollouts from the candidate list. For "no derived prefs", we remove the preference derivation part of the learning. Note that the preference pairs generated during the sorting process are still added to the dataset. Finally, "no ranking" means that the algorithm still selects preference pairs with an information gain acquisition function, but does not maintain a preference chain (so there are also no transitively derived preferences). This is simply the information gain algorithm of [2].

# E    Training Hyperparameters

The preference learning hyperparameters:

| Hyperparameter | Value |
|---|---|
| All algorithms | |
| Optimizer | Adam |
| Learning Rate | 1e-4 |
| Weight Decay | 3e-5 |
| Batch Size | 32 |
| Temperature | 0.1 |
| Fragment Length | 30 |
| Training Epochs | 20 |
| IMPEC | |
| Max. Preference Chain Size | 30 |
| Stop querying at | Epoch 15 |
| M | 10 |

The PPO training hyperparameters:

| Hyperparameter | Value |
|---|---|
| Training steps | 500,000 |
| Learning rate | 0.0017 |
| Gamma | 0.98 |
| Lambda | 0.975 |
| Entropy Coefficient | 0.15 |
| Batch size | 64 |
| Clip range | 0.2 |

# F  Ablation Studies

To understand what leads to IMPEC's performance, we experiment with removing three different components: (1) the active learning process; (2) preference derivations; and (3) the ranking process. The results can be found in Table 3.

We conduct each ablation experiment with 25 seeds and report the number of failed runs for each algorithm. A run fails when its final policy's average return $\leq 10$.

Our results suggest that there is no one component that is responsible for the entire gap between IMPEC and the baseline. However, the combination of ranking and active learning can be quite powerful: comparing the "no derived prefs" and the baseline, the failure rate immediately dropped by 50%.

Table 3: The failure percentages and their corresponding p-values of being significantly lower than the failure rate of the baseline

| ALGORITHM-VARIANT | FAIL % | P-VALUE (F%<BASELINE) |
|---|---|---|
| IMPEC | 2/25 | 0.082 |
| NO ACTIVE LEARNING | 4/25 | 0.267 |
| NO DERIVED PREFS, | 3/25 | 0.161 |
| & NO RANKING | 6/25 | 0.557 |
| BASELINE | 6/25 | - |

# G  Graph-theoretical approach

**The Preference Datasets**  We visualize the preference datasets gathered by the baseline and IMPEC on Lava-Position in Figure 5. The baseline dataset is randomly sampled at the start of the training, while we take a snapshot of the IMPEC dataset (which is constructed iteratively) at its last training epoch. The datasets are visualized as graphs, with each node being a unique rollout, and each edge representing a preference label. Both IMPEC and the baseline have a query budget of 600 pairwise queries, and IMPEC uses the first 400 of its queries to do pairwise comparisons between randomly selected rollouts, as opposed to actively selected rollouts. The IMPEC and baseline datasets have 405 and 538 unique rollouts as well as 791 and 594 unique edges, respectively. We include several other graph statistics in Table 4.

IMPEC and the baseline exhibit three notable differences in the graph properties. The first is their clustering coefficient, which measures the degree to which nodes tend to cluster together. The IMPEC graph has a higher clustering coefficient because of the many preferences it derives from the ranked chain. This is relevant to the number of chains in the graph: since most nodes are connected to IMPEC's central cluster through some edges, it creates many more chains between two nodes across the graph. In both graphs, we also observe that there are many chains of length 1 that are not connected to any larger cluster. We suspect that the algorithms' sample efficiency can be further improved if these pairs can be meaningfully linked to the clusters.

Finally, we measure the graph's efficiency, which is a metric from network science that is intended to measure the flow of information between "communicating" nodes [11]. The assumption which underlies the graph efficiency metric is that more distant nodes are less efficient at information exchange. To avoid inflating the metric with the many length 1 that are not connected to the rest

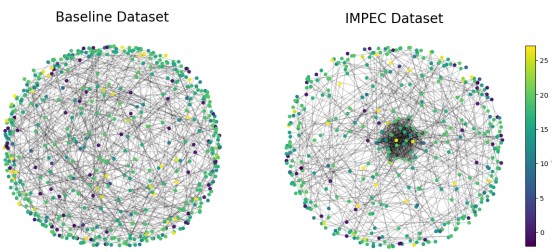

Figure 5: The preference datasets visualized as graphs. The IMPEC dataset connects more scattered rollouts through its central cluster.

|  | IMPEC | Baseline |
|---|---|---|
| Num. Nodes | 405 | 538 |
| Num. Edges | 791 | 594 |
| Cluster Coef. | 0.0642 | 0.0013 |
| Num. Chains | 416 | 81 |
| Efficiency | 0.19 | 0.11 |

Table 4: Properties of the dataset graph.

of the graph, we compute efficiency only on the two graphs' biggest connected components. We empirically observe that the IMPEC dataset graph has a higher average global efficiency than the baseline graph, which means that the average shortest path between vertex pairs in IMPEC is shorter than for the baseline. This raises an interesting question: Is the assumption in network theory, where the distance of nodes influences information efficiency, also applicable to preference learning? We do not have matured results establishing connections between data connectivity in RLHF and the training performance yet, but it would be an interesting next step for research.

## H  The Complete P-Value Table

Table 5: A complete list of all p-values of the algorithms performing better than the baseline for all tasks

|  | IMPEC | Information Gain | Volume Removal |
| --- | --- | --- | --- |
| Empty | 0.10 | 0.82 | 0.37 |
| Dynamic Obstacles | 0.09 | 0.66 | 0.77 |
| Lava | 0.08 | 0.26 | 0.44 |
| Lava Position | 0.01 | 0.46 | 0.20 |
| Fetch | 0.26 | 0.65 | 0.45 |
| Go To Door | 0.37 | 0.55 | 0.39 |

## I  Baseline Comparison and Data Scaling

We expect that the baseline should suffer less from reward confusion if given more comparison data. To test this hypothesis, fig. 6 shows return for the baseline with different query budgets, along with IMPEC results from our main experiments (which used 150 preference pairs and 300 queries). We tested the baseline with 6× the data used by IMPEC (i.e., 900 preference pairs and queries), then gradually pushed up the amount to 1500 (10× of IMPEC data). With 6× the data, we still see some learning failures, and the seeds' standard deviation decreases to IMPEC's level only when we use 10× data.

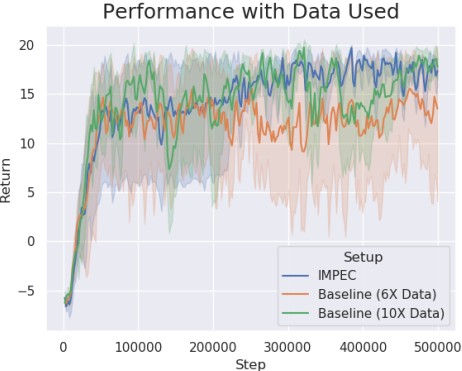

Figure 6: The learning curves for IMPEC and baseline with different amounts of data.

 **J    Learning Curves for All Tasks**

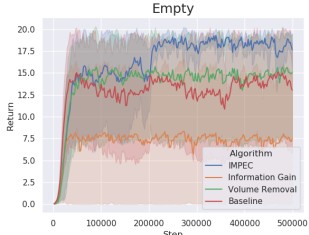
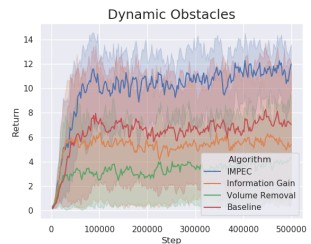
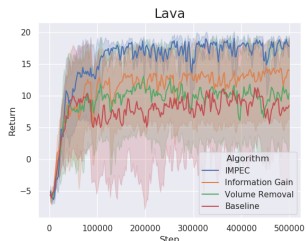

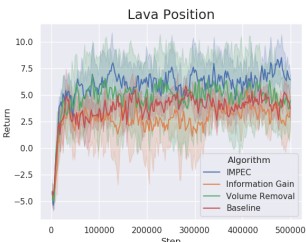
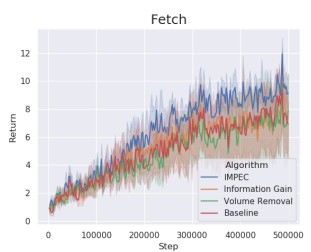
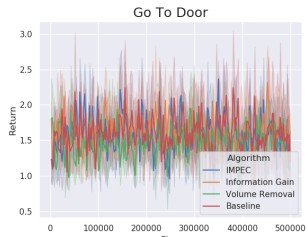

