# OpenReview forum: "Exploring and Addressing Reward Confusion in Offline Preference Learning"
_NeurIPS.cc/2024/Workshop/BDU — NeurIPS BDU Workshop 2024 Poster_

### Official Review · Reviewer_aMXA · 2024-09-26
**This paper presents a benchmarking environment for evaluating spurious correlations in Reinforcement Learning from Human Feedback algorithms and introduces an algorithm to address reward confusion. While the authors conducted a thorough evaluation, I am not convinced that the results directly support their claims regarding the algorithm's effectiveness in addressing reward confusion. I've listed my main comment below, I hope the authors find it helpful.**

**Rating:** 5
**Confidence:** 3

**Review:**

1. The authors propose Information Guided Preference Chain (IMPEC) to address reward confusion. However, the results in Table 1 do not definitively demonstrate its effectiveness. While IMPEC-trained agents outperform other approaches, it's unclear how this directly addresses the challenge of reward confusion. In the Confusing Minigrid (CMG) scenarios, spurious behavior is introduced by adding an extra state. The neural network might be able to ignore this additional state or assign it a lower importance in action determination, especially if it's just one component of the input state vector. A more targeted analysis could involve increasing the amount of spurious information in the CMG environment to rigorously evaluate the robustness of the proposed RLHF approach.

2. I commend the authors for their comprehensive analysis, including ablation studies and graph-theoretical evaluation.

---

### Official Review · Reviewer_zcrp · 2024-09-27
**Neat idea, but the algorithm is not described precisely.**

**Rating:** 6
**Confidence:** 2

**Review:**

## Strengths

The authors present an interesting approach to enforcing transitivity of
preferences during reward learning. To the best of my knowledge, this technique
is novel, and will be of interest to a large community. The authors also propose
the use of a Bayesian neural network to rank agent trajectories by information
gain, so as to construct an &ldquo;efficient&rdquo; set of trajectories for preference
learning. I really enjoyed their graph-theoretic analysis of the resulting
preference dataset in the Appendix&#x2014;this would have been nice to include in the
main text.

The authors also introduce a notion of *reward confusion*, which is essentially
describing the difficulty of learning causal relationships in preference
learning. Their hypothesis is that learning the full ordered chain of
preferences can help alleviate this problem. While I am
not totally sold that this should solve the problem, I think it is plausible
enough to warrant further investigation.


## Weaknesses

While I believe I understand the general idea, the text does not provide precise
details about how the IMPEC algorithm works. Notably, the provided pseudocode
references subroutines such as &ldquo;SupervisedTrain&rdquo; and &ldquo;DerivePreferences&rdquo; which
are not described anywhere. While the former is fairly self-explanatory, the
latter is not (to me). More importantly, the variable names used in the
pseudocode do not align with those described in the text, making it difficult to
follow. If I had to implement IMPEC right now, I would be mostly lost.


## Minor Issues and Questions

Typo on line 28, &ldquo;one other our major contributions&rdquo;, should say &ldquo;one of our
other major contributions&rdquo;.

The notation $\xi = (s_t, a_t)$ for a rollout is inaccurate &#x2013; $(s_t, a_t)$ would
generally be interpreted as a tuple, you really want $(s_t, a_t)_{t\ge 0}$.

There is a typo in the &ldquo;type signature&rdquo; of $\mathcal{R}$ on line 57, it says $\mathcal{R}:\mathcal{S}\times\mathcal{A}\to\mathcal{R}$,
where the latter $\mathcal{R}$ should probably be $\mathbb{R}$.

---

### Decision · Program_Chairs · 2024-10-09

Accept (Poster)